# Estimating Tsunami Economic Losses of Okinawa Island with Multi-Regional-Input-Output Modeling

**Kwanchai Pakoksung [1],\*, Anawat Suppasri [1], Kazuyo Matsubae [2] and Fumihiko Imamura [1]**

[1]   International Research Institute of Disaster Science, Tohoku University, 468-1 Aramaki-Aza, Aoba-ku, Sendai 980-0845, Japan

[2]   Environmental and Energy Economics Laboratory, Division of Environment Studies, Department of Environmental Studies for Advanced Society, Graduate School of Environmental Studies, Tohoku University, 468-1 Aramaki-Aza, Aoba-ku, Sendai 980-0845, Japan

\*   Correspondence: pakoksung@irides.tohoku.ac.jp; Tel.: +81-22-752-2090

**Abstract:** Understanding the impacts of tsunamis, especially in terms of damage and losses, is important for disaster mitigation and management. The aim of this study is to present our estimations of the potential losses from tsunami damage on Okinawa Island. We combine the use of a tsunami hazard map and our proposed economic loss model to estimate the potential losses that would be sustained by Okinawa Island in the event of a tsunami. First, to produce the tsunami hazard map, we calculated tsunami flow characteristics using the mathematical model TUNAMI-N2 and incorporating 6 earthquake fault scenarios around the study area. The earthquake scenarios are based on historical records along the Ryukyu Trench and the Okinawa. The resulting inundation map is overlaid with economic land use type and topography maps to identify vulnerable regions, which are then employed to compute potential economic losses. Second, we used our proposed economic model for this study area to calculate the potential losses that would be sustained in these vulnerable regions. Our economic model extends the multi-regional-input-output (MRIO) model, where the economic values of industrial sectors are scaled to correlate with land use and topography types (coastal and inland areas) to calculate losses through the Chenery–Moses estimation method. Direct losses can be estimated from the total input of the MRIO table, while indirect losses are computed from the direct losses and interaction parameter of the MRIO table. The interaction parameter is formed by linear programming and calculated using the Leontief methodology. Our results show that the maximum total damaged area under the 6 earthquake scenarios is approximately 30 km$^2$. Inundation ranging from 2.0 to 5.0 m in depth covers the largest area of approximately 10 km$^2$ and is followed by areas with inundation depths of 1.0–2.0 m and >5.0 m. Our findings show that direct losses will occur, while indirect losses are only approximately 56% that of direct losses. This approach could be applied to other areas and tsunami scenarios, which will aid disaster management and adaptation policies.

**Keywords:** tsunami hazard; tsunami modeling; tsunami economic losses; Input-Output table

## 1. Introduction

Tsunami, one of Earth's major natural hazards, affects human life and property. Inundation caused by tsunami flooding often affects agricultural, industrial, and urban areas. The impacts of tsunami floods are expected to intensify in the coming years due to population growth, population migration to coastal areas, and climate change [1]. In the case of Japan, the agricultural, industrial, and commercial sectors, which are crucial for global exports, were seriously damaged by tsunami floods from the 2011 Great East Japan earthquake. The 2011 flood affected many urbanized areas (which consist of commercial and industrial sectors) within 300 kilometers along the East Japan coast, and the total

damage amounted to approximately USD 30,000 million [2–9]. The results of natural disasters indicate the need to develop effective mitigation strategies after the 2011 Great East Japan earthquake.

### 1.1. Tsunami Model

Tsunami hazard assessment is usually estimated with numerical modeling based on an individual scenario. Currently, there are several tsunami models for estimating tsunami damage, such as the Cornell Multi-Grid Coupled Tsunami Model (COMCOT) [10,11], Method of Splitting Tsunami (MOST) [12], and Tohoku University's Numerical Analysis Model for Investigation of Near-Field Tsunami, Number 2 (TUNAMI-N2) [13]. These models are based on the non-linear shallow water equation. Several studies on tsunami damage have been previously performed. For example, the Arabian Sea tsunami, generated by the 1945 Makran earthquake, has been simulated by the TUNAMI-N2 model to assess the tsunami hazard in the area [14], which is based on several historical earthquake hypotheses [15]. The tsunami hazards of the Car Nicobar coast have also driven the use of TUNAMI-N2. A study was performed on the 2004 Indian Ocean tsunami to understand the effects of wave run-up in the Koodankulam region of the Tamil Nadu coast using the TUNAMI-N2 model. The 2004 tsunami was simulated using 28 scenarios with a run-up wave range between 1.30 and 3.54 m [16]. In addition, the 2004 tsunami damage in Thailand was estimated using numerical modeling and satellite remote sensing. The modeling of the tsunami damage provided by the TUNAMI-N2 model demonstrated reasonable accuracy [17,18]. The TUNAMI-N2 model has been applied to the 2011 Tohoku-oki tsunami to understand inundation characteristics and to identify the tsunami hazard in Sendai city [19,20]. The results show that the flood depth ranges between 2.4 and 6.0 m, and the flow velocity ranges between 3.4 and 6.2 m/s. For the 2011 Tohoku-Oki Earthquake and Tsunami, the direct economic impact was estimated by insurance documents as approximately 2.3 billion USD [21,22]. This study required economic losses from other sources, such as physical damage losses and manufacturing losses from the supply chain.

### 1.2. Economic Model

Damage cost estimation can influence the decision-making process in disaster management and, in turn, affect the effectiveness of mitigation strategies. Including damage cost estimations in the decision-making process could improve the efficiency of mitigation strategies. Damage cost estimates can inform planners of the types of mitigation strategies that can be implemented to reduce losses. Losses in a tsunami event are not limited to the direct losses sustained from physical damage (e.g., buildings and loss of lives) in a flood, it is also important to account for indirect economic losses that are likely to arise when operations are unable to return to normalcy. Indirect losses also vary in space as the various sectors are interlinked and have spatial divisions.

The input-output (IO) model is one of the most appropriate methods for estimating these impacts [23]. IO tables describe the costs of production for each product and in each region. IO tables also link multiple regions at the same time, describing the transaction relationships of goods. The IO table covers only domestic trades, but the combination of multiple IO models can provide a more accurate representation of the monetary flow between countries for a specific product. The application of the IO table has been widely applied. The water transfer pattern was analyzed by linking the California and the Arizona IO models [24]. A modified IO model was developed to qualify the effects of structurally-related factors in water use practices on industrial water use [25]. The effects of the inter-regional industry on water demand are not only from product processes but also from other industries [26]. The water footprint was analyzed in the Kanto Basin using a Kanto inter-regional IO table [27]. The Inter-Regional Input-Output (IRIO) table was applied by estimating tsunami economic losses on Okinawa Island [23]. The study concluded that tsunami losses were limited by overestimation because the damaged area was specific to some regions, whereas the industrial sector used to estimate the losses from the IRIO table represents the economic value of a whole region. Additionally, economic losses were not the only area of interest (coastal area) that affected other regions

with economic linkages. The study recommended that the IRIO table be downscaled to the specific area as the disaster area.

The combination of multiple IO models or downscaling IRIO is also known as the multi-regional input-output (MRIO) model [27]. The amount of goods and services produced in each region is the final demand, which includes the consumers' demands both regionally and internationally. The MRIO table and damage cost analysis are powerful valuation tools that use the economic values of each industrial sector in the impacted disaster area to evaluate the direct losses and the indirect losses that are associated with other parts of a demand/supply chain and internationally traded products [28].

It is well known that the MRIO table is mostly based on two types, Isard-type and Chenery–Moses-type [29,30]. These two types contrast in the assumption of the interregional trade coefficient. The former involves a complete set of intrastation and interregional data, making it difficult to obtain such interregional trade coefficients for each sector in each region, especially within a country. Conversely, the latter has advantages in accumulating the table because of the simplification in applying the common interregional trade coefficient for each sector in each region. Thus, the MRIO table within a country is usually compiled as Chenery–Moses-type. However, the interregional trade data within a country are still difficult to obtain completely, which forces one to shift from adopting the survey method to some kind of non-survey method in the estimation of the interregional trade coefficients. RAS method is usually used to estimate the coefficient. The RAS method is used for estimating the input output coefficients, for which only the peripheral information of the column sums and the row sums is known, in an iterative way. This method requires initial values upon which the solution depends [31].

A review of the literature shows that there are few demonstrations of the application of the IRIO table for evaluating tsunami economic losses in each economic product, such as agriculture, industry, and commercial, for a number of reasons [27]. It is necessary to adapt the MRIO table for tsunami loss estimation. In terms of disaster damage cost assessment, the MRIO table has the potential to estimate the damage cost, but it still produces an approximate value. An approximate value is obtained from the MRIO table because a sector in the table is used to represent an entire region. While the disaster occurs in some areas of the region, it is not appropriate to apply the MRIO table to conduct the damage loss assessment directly. For a tsunami disaster, the damage area is usually in a coastal area where the area has a flat slope and closes to the coastline.

### 1.3. Objective of this Study

The objective of this study is to estimate the maximum potential damage losses, both direct and indirect, of Okinawa Island, Japan in a tsunami event. We begin the study by producing a tsunami inundation map, using the TUNAMI-N2 model to identify the extent and areas of inundation. We adopt the IRIO table provided by the Ministry of Economy, Trade and Industry (METI) in Japan [32] for the estimation of tsunami losses. Then, we develop an MRIO table for Okinawa Island by downscaling the IRIO table by geographic characteristics to divide the economic value between the coastal and inland areas. Finally, the losses are estimated by using the economic structure presented in the MRIO table. Direct losses can be derived from the MRIO table, while indirect losses can be estimated by using both the direct loss estimates and the transactions parameter of the MRIO table to consider the transactions among regions country and economic sectors.

Okinawa Island, in the southern part of Japan, as shown in Figure 1, the island is very important for tourism, with the number of travelers increasing by 2%–10% in the last 10 years. Tourism causes economic growth the Okinawa region. However, natural disasters, such as tsunamis, can occur in this region. Figure 1a presents the historical earthquake record around Okinawa Island [33,34], and some of these earthquakes have caused tsunami damage on Okinawa Island. Two important events are shown in Figure 1a: the M7.4 1771 large tsunami close to Ishigaki Island and the 1791 earthquake near Okinawa Island in the south [35]. The 1771 event occurred offshore of Ishigaki Island approximately 500 km from the study area. The estimated initial flow depth of the tsunami was approximately 80 m [36]. The M8.2

1791 tsunami occurred along the Ryukyu Trench approximately 50 km from the study area. This event had a recorded flow depth of approximately 1.5–11.0 m [35], as shown in Figure 1b.

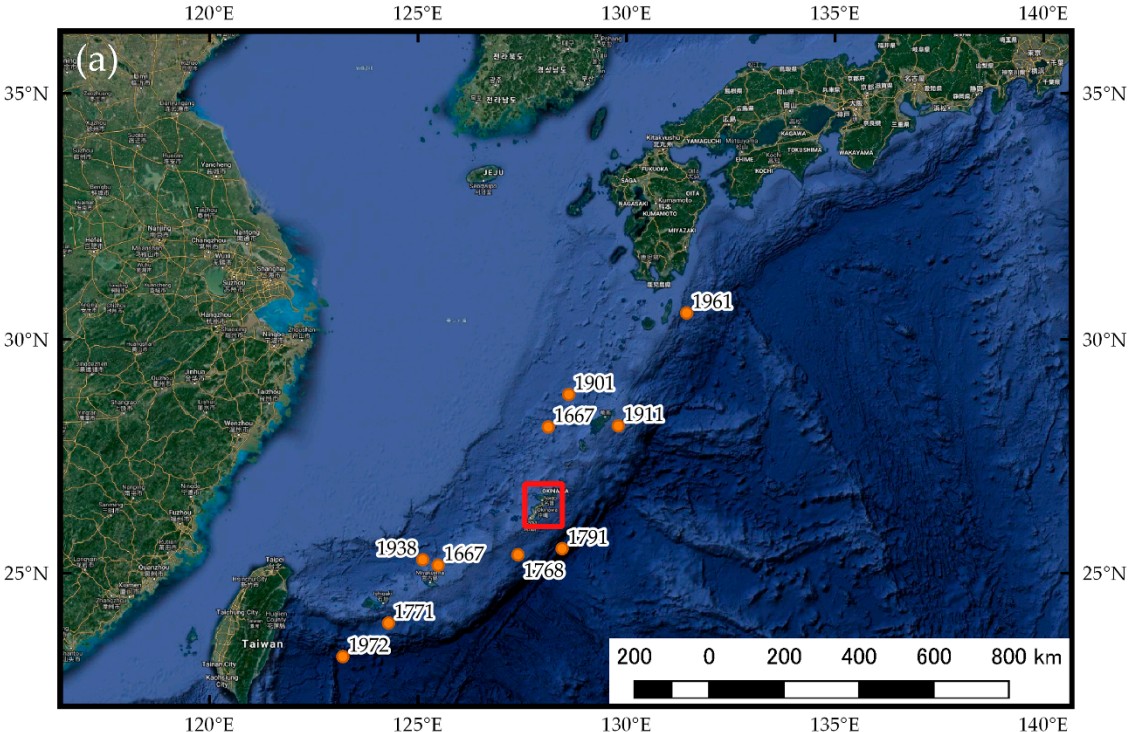

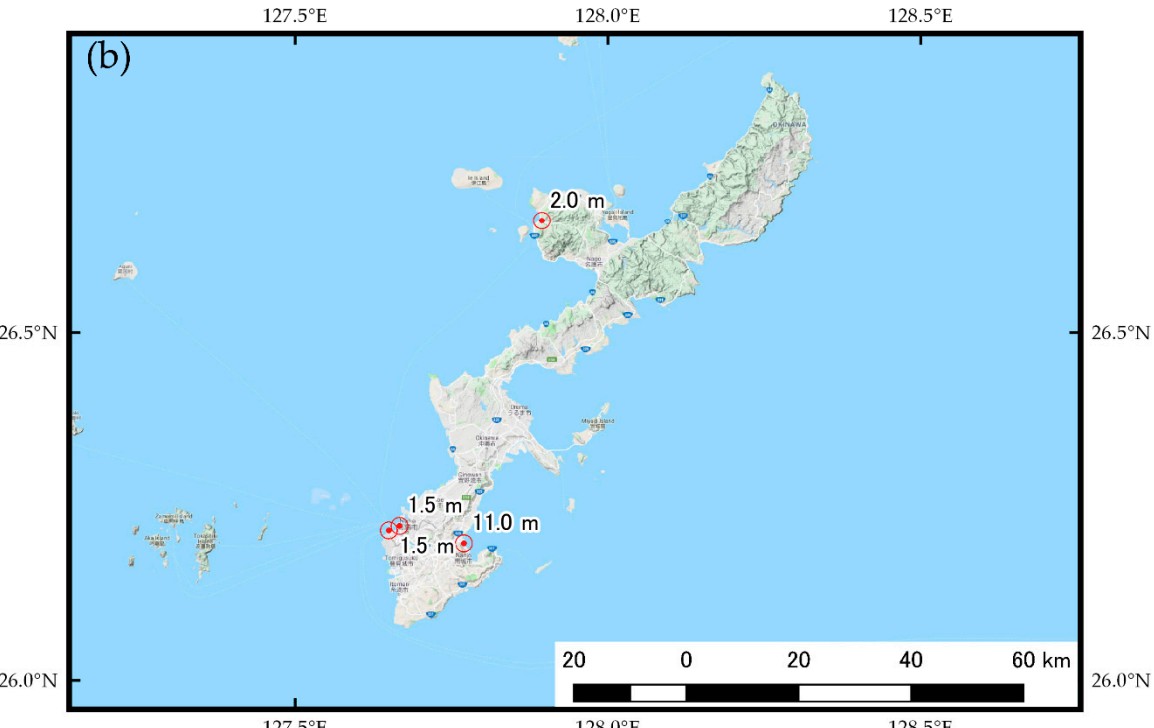

**Figure 1.** Location of Okinawa Island, (**a**) Study area in the red box and historical earthquakes around Okinawa Island, (**b**) The observation data on the historical event (1791).

## 2. Materials and Methods

There are two components to this study: the tsunami simulation model and the economic losses model, as shown in Figure 2. First, a tsunami hazard map is generated based on earthquake scenarios around the study area and computed by a mathematic model (TUNAMI-N2). We perform the simulation based on 6 earthquake scenarios along the Ryukyu Trench and the Okinawa Trough around Okinawa Island. TUNAMI-N2. The resulting hazard map is overlaid with a land use and topography map to identify vulnerable regions for computing economic losses. The economic model of this study area follows the MRIO model, where industrial sectors are scaled to correlate with the economic land use type and topography type (coastal area and inland area) by the Chenery–Moses-type. The relationship between the hazard map and the economic values is used to estimate the direct and indirect disaster losses. Direct losses can be directly estimated from the total income of the MRIO table, while the indirect losses can be computed by the direct losses and the interaction parameter of the MRIO table. The interaction parameter is formed in linear programming and is calculated using the Leontief methodology.

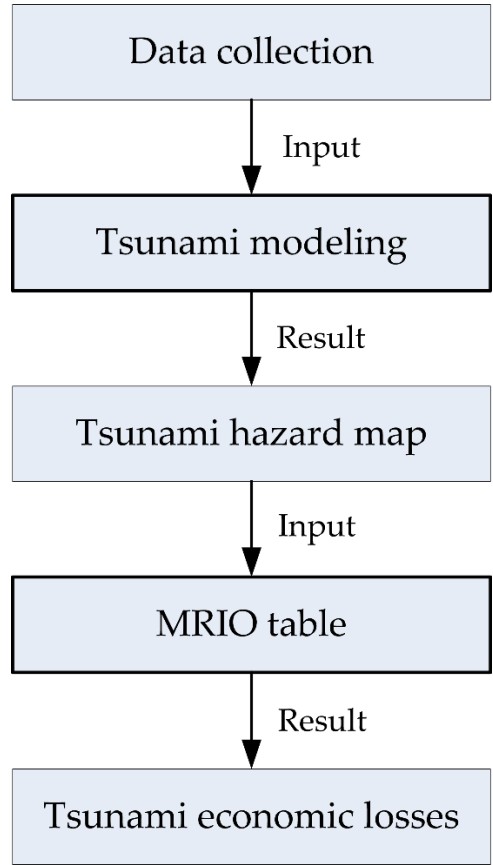

**Figure 2.** Stream line of this study used to achieve tsunami economic losses.

### 2.1. Tsunami Source Model from Earthquake Fault Scenario

An important variable for generating the initial water level of a tsunami is the seafloor deformation, which has been computed from fault modeling, as the tsunami source model. Because of the complexities and uncertainties of fault rupture processes, the initial water level estimated in this study is based on a rectangular fault model, and we assume that the change in sea surface is the same as the seafloor deformation. The fault model also assumes that the fast movement of the sea surface is the only vertical displacement that occurs [37]. The fault parameter of the Ryukyu Trench and the Okinawa Trough earthquake is proposed by the Okinawa Prefectural Government [35]. In addition, several studies have

shown that constant slip faults are associated with smaller tsunami, compared to those simulated with more realistic spatially varying slip [38–40]. The fault model is established by 6 earthquake scenarios based on the 1791 tsunami event in the Ryukyu Trench [35]. The locations of the 6 fault scenarios are presented in Figure 3. The fault parameter information is shown in Table 1.

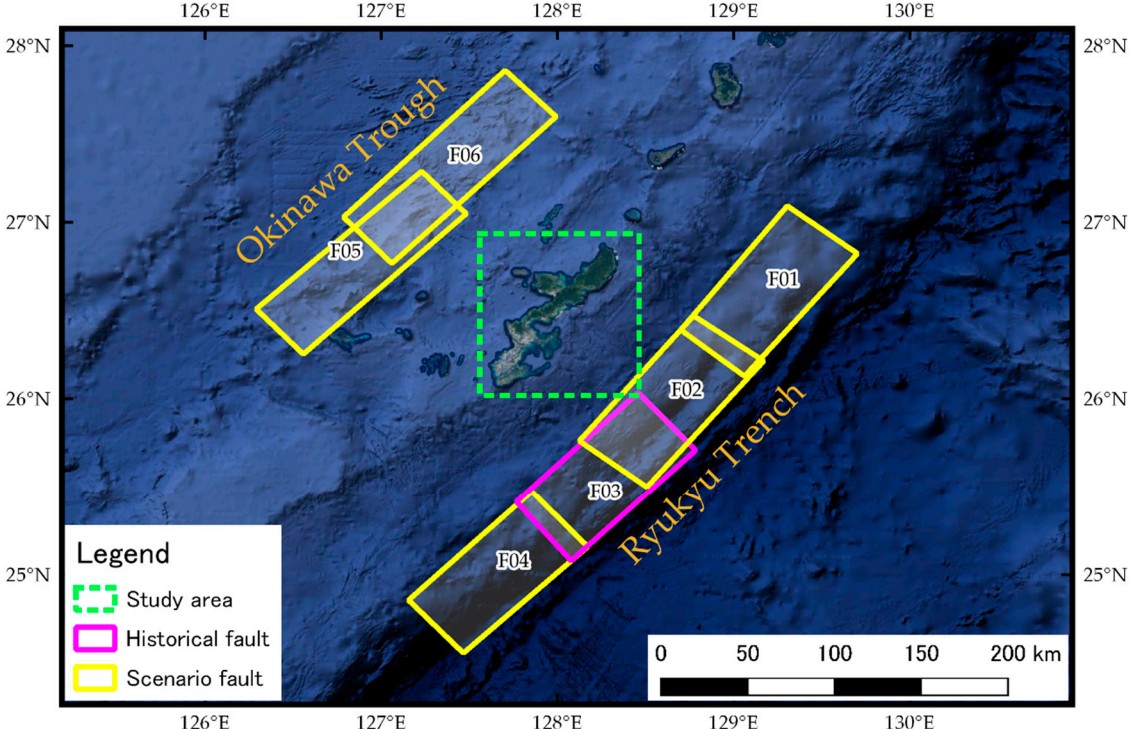

**Figure 3.** The fault scenario around Okinawa Island that was selected to evaluate the tsunami hazard. The study area is in the green box, which is represented by the 90 m grid size for the inundation computation.

**Table 1.** Fault parameters for earthquake scenarios used to generate tsunami.

| No | Name | Lat. | Lon. | Width, km | Length, km | Depth, km | Strike, deg. | Dip, deg. | Rake, deg. | Slip, m | Mw |
|----|------|------|------|-----------|------------|-----------|--------------|-----------|------------|---------|-----|
| 1 | F01 | 26.812 | 129.756 | 100 | 50 | 5 | 218 | 12 | 90 | 12 | 8.2 |
| 2 | F02 | 26.196 | 129.172 | 100 | 50 | 5 | 218 | 12 | 90 | 12 | 8.2 |
| 3 | F03 * | 25.728 | 128.806 | 100 | 50 | 5 | 225 | 12 | 90 | 12 | 8.2 |
| 4 | F04 | 25.181 | 128.163 | 100 | 50 | 5 | 225 | 12 | 90 | 12 | 8.2 |
| 5 | F05 | 27.126 | 127.519 | 130 | 40 | 2 | 225 | 30 | 270 | 8 | 8.1 |
| 6 | F06 | 27.650 | 128.050 | 130 | 40 | 2 | 225 | 30 | 270 | 8 | 8.1 |

Remark * This fault based on the historical event in 1791.

### 2.2. Tsunami Modeling

To obtain the tsunami inundation for different earthquake scenarios, a numerical tsunami simulation is driven using the TUNAMI-N2 model [5]. It is run on a computational domain using a nesting grid system from larger areas to smaller areas to cover Okinawa Island. The model assumes the nonlinear theory of the shallow water equation solved by the leap-frog scheme to model tsunami propagation and inundation on terrain. A finite difference methodology is applied to run the nonlinear equation. The bottom friction is represented by Manning's roughness coefficient of 0.025 [41]. The nonlinear system is solved at each time step of 0.01 sec. At the boundary lines, the open

sea represents a limit with non-reflective boundary conditions, and coastal areas have no specific boundary conditions for wet/dry fronts [13].

A bathymetric grid is prepared for tsunami propagation and inundation simulation. The tsunami modeling grid is divided into three regions, 810 m, 270 m, and 90 m based on data provided by the Geospatial Information Authority of Japan (GSI) [2]. The flow depth resulting from the tsunami model is used to identify the tsunami flood map for the economic loss estimation. In the topography of the flood map, a flow depth greater than 0.3 m is used as the threshold value between flood and non-flood areas. The threshold value of 0.3 m is identified by the ground floor height of the building in the urban area and the normal water depth for the agricultural area in planting.

### 2.3. Multi-Regional-Input-Output Table

A single-regional input-output (SRIO) table, as a conventional input-output (IO) table, can generally be used to estimate direct and indirect economic effects by examining the economic linkages between sectors and regions [42]. The SRIO table is explained in a monetary matrix (row and column) format containing inter-industry transactions. The rows of the matrix describe the distribution of outputs (product sale structure). The columns display inputs (purchase structure), the sum of raw materials and the value-added expenses (for details, see [43,44]). The equation used in the SRIO model with the mixed variables has been derived as follows. First, the vector of output is the sum of the intermediate transactions and the final demand (1); then, Equation (1) is transformed into a matrix as shown in Equation (3), as illustrated by the basic equation for IO tables with the Leontief inverse matrix.

$$x = A \cdot x + f \tag{1}$$

$$(I - A) \cdot x = f \tag{2}$$

$$x = (I - A)^{-1} \cdot f = L \cdot f \tag{3}$$

where, $x$ is the vector of output, $f$ is the vector of final demand, $A$ is the matrix of input coefficient, $I$ is the identity matrix, and $L$ is the Leontief inverse matrix $(I - A)^{-1}$ as $(L_{ij})$. When three sectors are assumed for the IO table, Equation (4) is as follows:

$$\begin{bmatrix} x_1 \\ x_2 \\ x_2 \end{bmatrix} = \begin{bmatrix} L_{11} & L_{12} & L_{13} \\ L_{21} & L_{22} & L_{23} \\ L_{31} & L_{32} & L_{33} \end{bmatrix} \cdot \begin{bmatrix} f_1 \\ f_2 \\ f_3 \end{bmatrix} \tag{4}$$

However, the conventional IO table as an SRIO table only demonstrates these processes in one region, thus, the impacts of domestic trade cannot be presented.

For estimating indirect losses, $d_1$ is the direct losses at sector 1 on $f_1$ while $f_2$ and $f_3$ are 0. In the Leontief inverse matrix, the coefficient of sector 1 is also changed for estimating the indirect losses of this sector as $\Delta x_1$. The other indirect losses, $x_2$ and $x_3$, affected by $d_1$, are investigated in Equation (5).

$$\begin{bmatrix} \Delta x_1 \\ x_2 \\ x_2 \end{bmatrix} = \begin{bmatrix} (L_{11} - 1) & L_{12} & L_{13} \\ L_{21} & L_{22} & L_{23} \\ L_{31} & L_{32} & L_{33} \end{bmatrix} \cdot \begin{bmatrix} d_1 \\ 0 \\ 0 \end{bmatrix} \tag{5}$$

The SRIO table provided by the government has a limited ability to evaluate disaster areas because the disaster has an effect on other regions with economic linkages [27]. In the case of Japan, there is mainly an economic network in 9 regions with 81 networks, and the economic value of the 81 networks is presented in Figure 4. The network economic links of the 9 regions, which have been provided by METI, can be identified as the IRIO table [30]. The METI IRIO table of Japan contains 53, 29, and 12 sector classifications.

**Figure 4.** Economic data in Japan represented by 9 large regions. Economic network based on the 9 regions in Japan and the trade value between regions.

The use of the IRIO table estimates the indirect losses from disaster, and the estimation has a limitation because the disaster does not occur in the whole country or regional area [23]. The implementation of IRIO for estimating disaster economic losses requires a downscaling of the industrial sector specific to the disaster area. The method for downscaling the IRIO of the region of interest specific to the disaster area is shown in Figure 5. First, selecting a value in the IRIO to assess the study area results in an SRIO with trade value. The industrial sector in the selected SRIO table has a relationship with the land use type, but the disaster area is related to characteristics of geography. For tsunamis, the disaster area occurs in a coastal area where the selected SRIO table must point to the coastal zone. Downscaling the selected SRIO table to the expected area of tsunami disaster, which is original to this research, can be done based on the characteristics of geography, e.g., the coastal and inland areas. In this study, we hypothesize that the coastal area, as the disaster zone, is the high-vulnerability area for tsunami effects. The coastal areas are identified based on the vulnerability class of tsunami hazards proposed by Sambah and Miura [45]. That study proposed vulnerability classes of tsunami hazards from low to high based on the 2011 large tsunami event in Japan. We used the topography characteristics of the medium class from the previous study to classify the coastal areas in this study on Okinawa Island. The criteria of the 3 characteristics of geography used to identify the coastal area are shown in Table 2. The criteria used to divide the coastal area include the 3 characteristics of geography: altitude, gradient and distance from the sea, as determined by the disaster zone map (see Figure 6a). The distance from the sea is the measurement length from the coastal line, and we hypothesize that the coastal area is closed to the coastal line. Based on the tsunami wave runup processes, we classified the high vulnerability of tsunami runup area and the low vulnerability of tsunami runup area using 2 variables, gradient, and altitude. The tsunami wave is

easy to runup in a low gradient, while the high gradient is difficult. In addition, tsunami runup is difficult for high elevations but easy for low elevations.

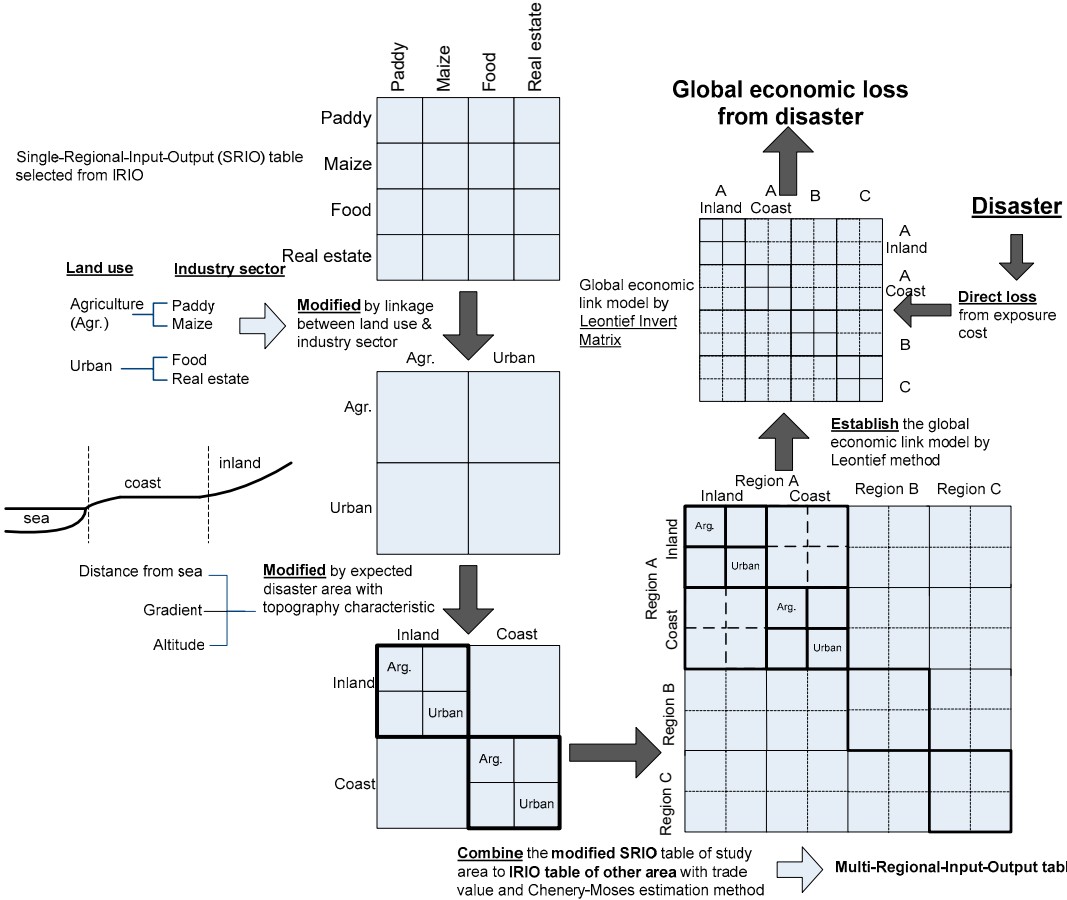

**Figure 5.** Method for constructing the multi-regional input-output (MRIO) table used to estimate the economic losses.

**Table 2.** Criteria of the 3 characteristics of topography used to identify the coastal zone.

| Topography Characteristic | Criteria |
|---|---|
| Distance from sea | <3.0 km |
| Gradient | <7.5 degree |
| Altitude | <20 m. MSL |

Remark: Mean sea level (MSL).

In the modification of the selected SRIO table, the ratio of land use (see Figure 6b) in the coastal area and inland area is used to separate the economic value in each sector into different zones because of the relationship between land use and sector in the SRIO table. For the estimated effect in other regions, the modified IRIO table is combined with the IRIO table of other regions by the Chenery–Moses estimation method (for details, see [46,47]) to establish the MRIO table. The MRIO table can be used to calculate the indirect loss by the Leontief method, as mentioned above. The table contains the economic link from the micro scale (disaster area) to the global scale. This paper uses the model based on Equation (5) to complete the MRIO table to calculate the indirect economic loss of tsunami flood disasters on industrial sector production.

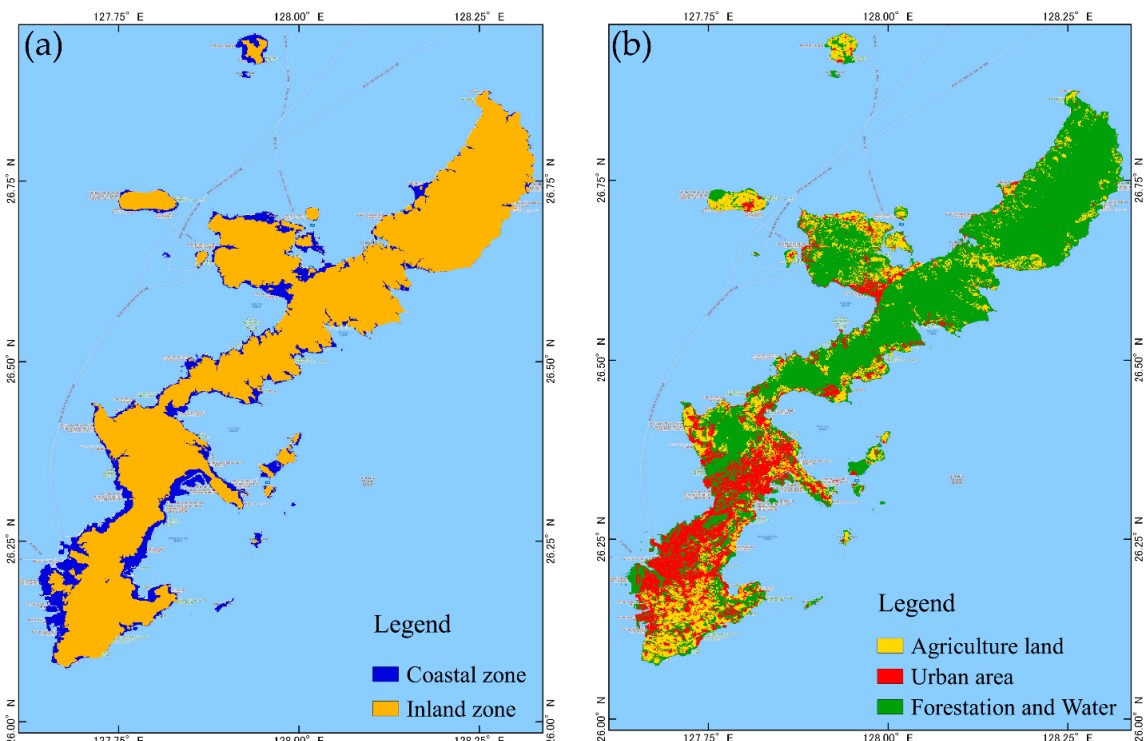

**Figure 6.** Parameters used to construct the MRIO, (**a**) is the topographical characteristic and (**b**) is the economic land use type.

## 3. Results and Discussion

### 3.1. Tsunami Flood Map

Figure 7 presents the initial water level distribution generated by the fault scenario model around Okinawa Island based on the Okada formula [31], where the maximum water level or uplift is approximately +5.0 m T.P., while the minimum water level or subsidence is −5.0 m T.P. The T.P. datum is the Tokyo Peil datum referenced by the sea water level in Tokyo Bay [2]. Four scenarios (F1, F2, F3, and F4) have the highest initial water levels among the six scenarios because of their slip and magnitude. Two scenarios (F5 and F6) show a lower initial water level than that of the four scenarios with the highest displacement area because of their fault size.

Figure 8 shows the inundation area from the TUNAMI-N2 model based on the 16 scenario faults. The damaged areas over the six faults is approximately 30 square kilometers. The inundation in the range of 2.0–5.0 m is the largest area of approximately 10 square kilometers, followed by that of 1.0–2.0 m and >5.0 m. In all scenarios, the inundation area is mostly located in the southern part of the island because of the topographical characteristic of urbanization. Our tsunami model results can be evaluated according to scenario F3 in Figure 8c. The comparison of flow depth between the observed and simulated scenarios point to point is presented in Table 3, which indicates an average difference of approximately 1.33 m. The F3 model results include $K$ of 1.12 and $\kappa$ of 1.03, which are in fairly good agreement with the observed data, as mentioned in Japan Society of Civil Engineering (JSCE) [6,17,18,20]. Figure 6 shows that the southern part of the island is a coastal zone (flat area and low altitude) that is easily affected by tsunami wave run-up. When the inundation results are overlaid with a land use map at the same location using the GIS technique though QGIS, it displays the estimated spatial inundation area in each land use type. Figure 9 presents the inundation area of each economic land use type in each scenario. The inundation damage in the urban area of the coastal zone occurred over an area of approximately 14.5 square kilometers, and this was the largest among

all land use types. The second greatest amount of damage occurred in the agricultural region of the coastal zone, and the least damage was in the urban area of the inland zone.

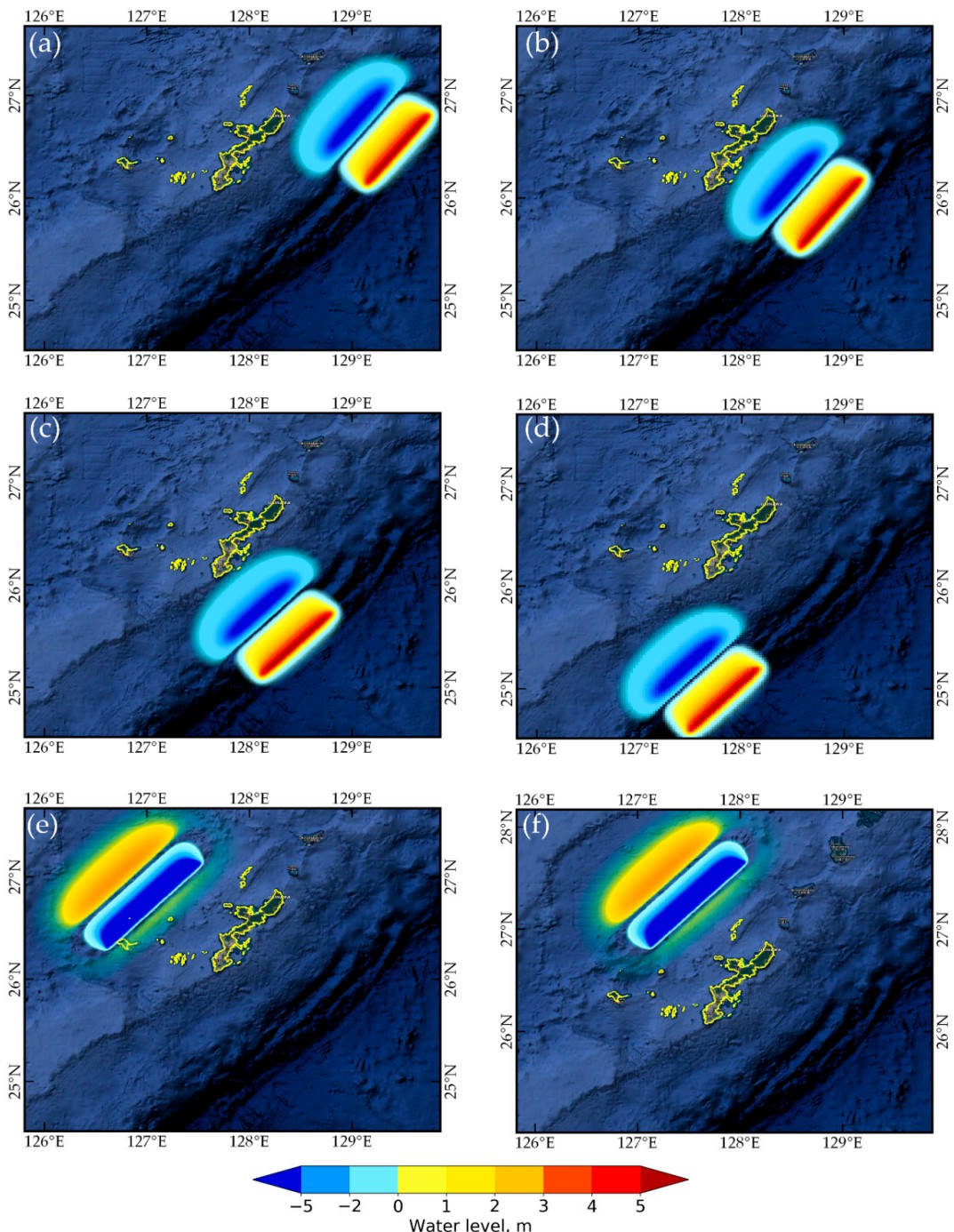

**Figure 7.** Initial water level generated by the fault parameter of the earthquake scenario. (**a**) F1, (**b**) F2, (**c**) F3, (**d**) F4, (**e**) F5, and (**f**) F6.

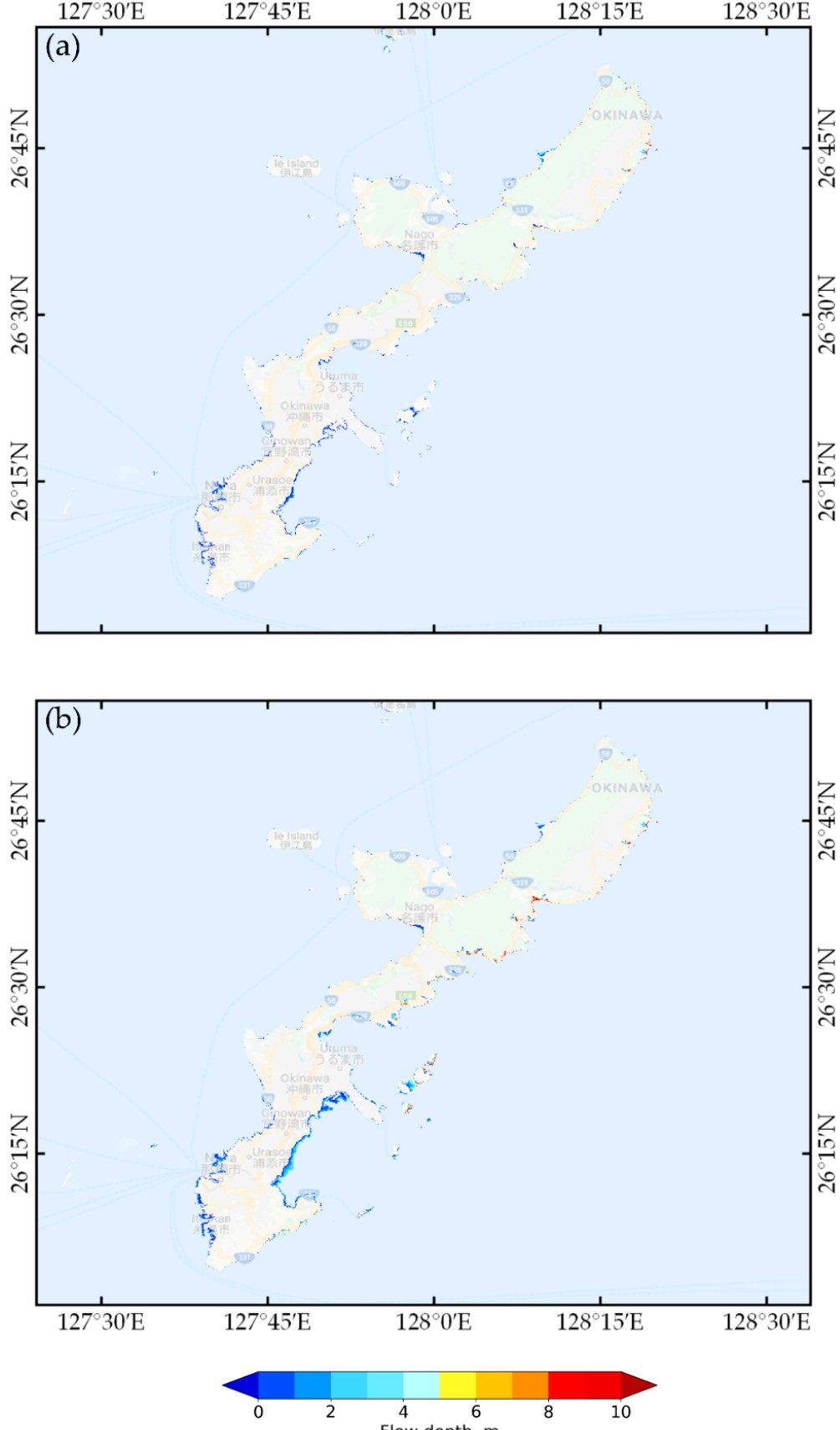

**Figure 8.** *Cont*.

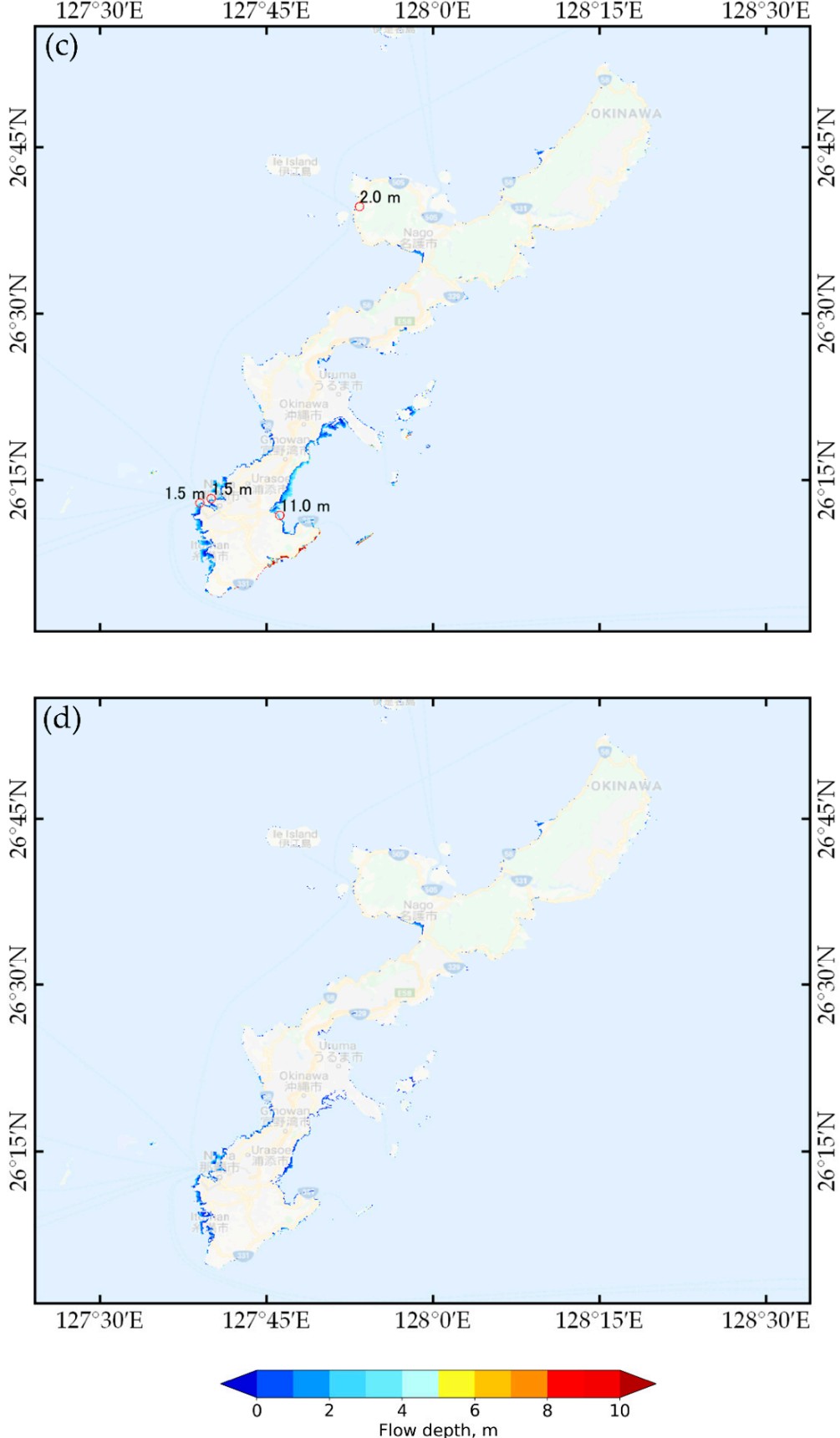

**Figure 8.** *Cont.*

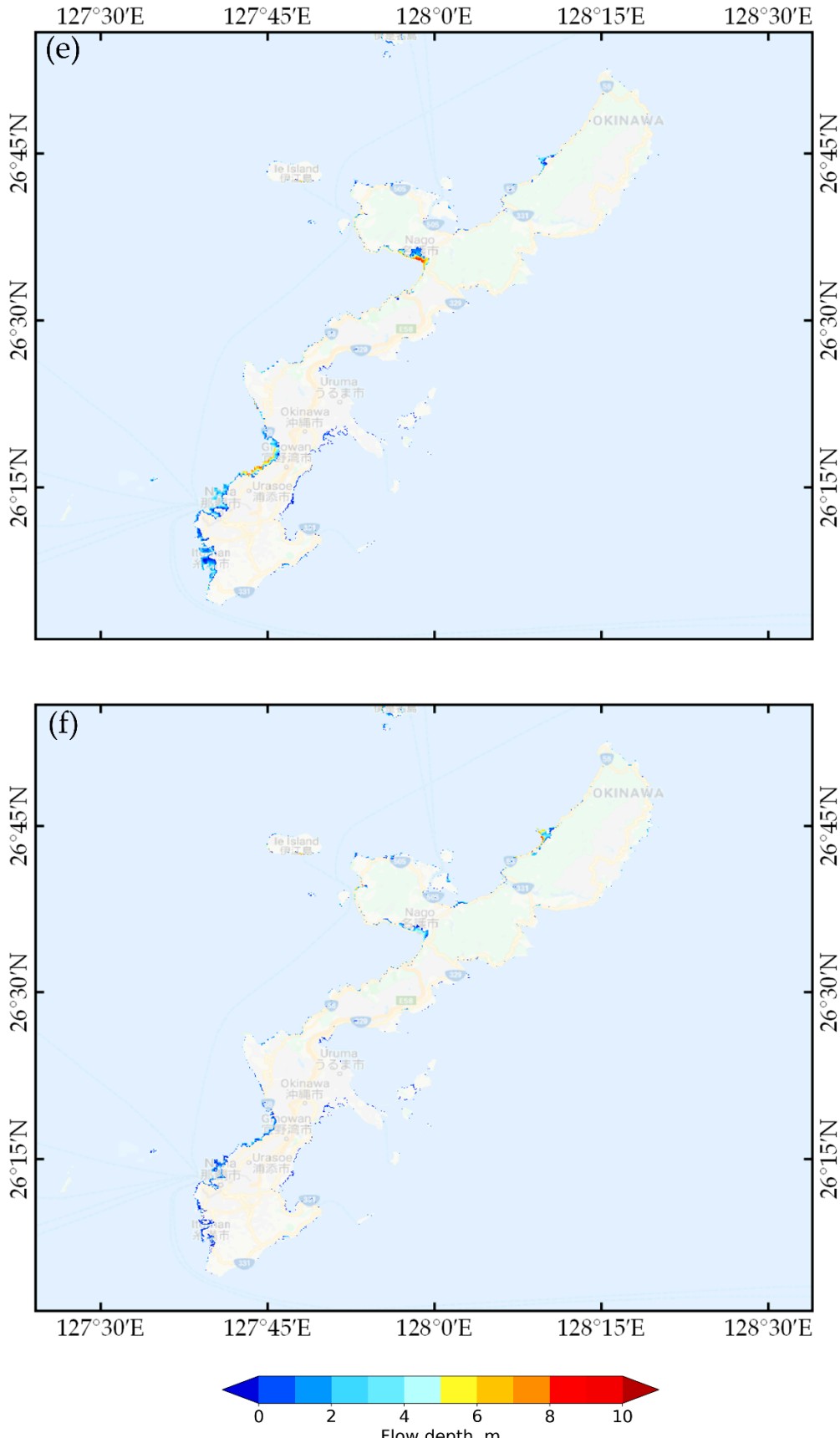

**Figure 8.** Maximum flow depth generated by the fault parameters of the earthquake scenario. (**a**) F1, (**b**) F2, (**c**) F3, (**d**) F4, (**e**) F5, and (**f**) F6.

**Table 3.** Comparison between observation data and simulation results.

| No | Observed Flow Depth, m | Simulated Flow Depth, m | Different, m |
|---|---|---|---|
| 1 | 1.5 | 1.62 | 0.12 |
| 2 | 1.5 | 1.70 | 0.20 |
| 3 | 2.0 | 1.80 | 0.20 |
| 4 | 11.0 | 6.20 | 4.80 |

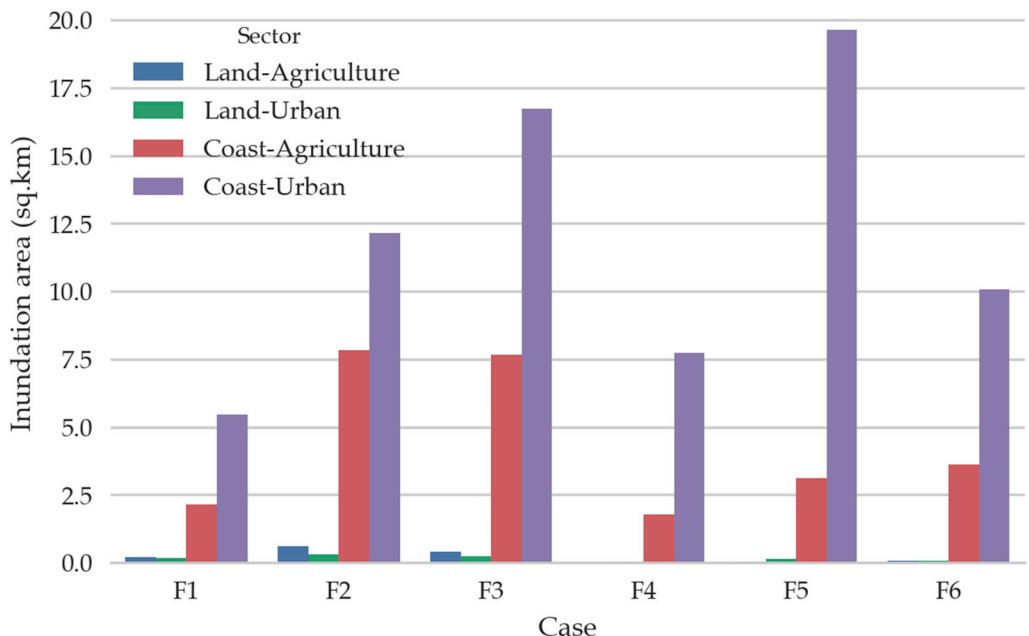

**Figure 9.** Inundation area presented for the economic sector in each tsunami case.

*3.2. Tsunami Economic Losses*

The MRIO table and tsunami losses on the economic product of the Japan region are calculated from traditional inter-regional data and numerical techniques. Figure 10 displays the MRIO table of Japan with two sectors, agriculture and urban (industrial and commercial), with a summary for 2005. In the MRIO table, the economic condition of the Okinawa region was divided into two small classes (coastal and inland areas) that presented the economic linkages with the other eight regions. There are three different parts: (1) aggregated MRIO table in a monetary unit, (2) land use area unit cost estimation with a unit of million USD per square kilometer, and (3) direct loss calculation with a unit of million USD. As seen in the MRIO model, the status of the regional economics of Japan is summarized for 2005. All input and output values in each region are related to a number of sectors as intermediate and total outputs. The total of each land use type area was used to calculate the area unit cost using the economic value in the MRIO table, as shown in Table 4. The cost value unit is usually based on Japanese YEN. For this study, the Japanese YEN was transferred to USD by using a rate of approximately 1 USD per 100 YEN. In this study, based on the MRIO table, the area unit cost of agricultural land in the Okinawa coastal zone is approximately 0.995 billion USD per square kilometer, while 29.55 billion USD per square kilometer is calculated for urban areas. The total flood area resulting from the previous process is approximately 19.0 square kilometers in the case of scenario F5. The area of tsunami damage was used to estimate the direct losses by using the area unit cost from a previous part of the MRIO table. For the production of Okinawa Island based on the tsunami disaster in 2005, the direct losses are presented in Figure 11a, which have an average value of approximately 350 billion USD. The direct losses in the urban area of the coastal zone are the highest losses among land use types because of the highest unit cost and largest inundation area.

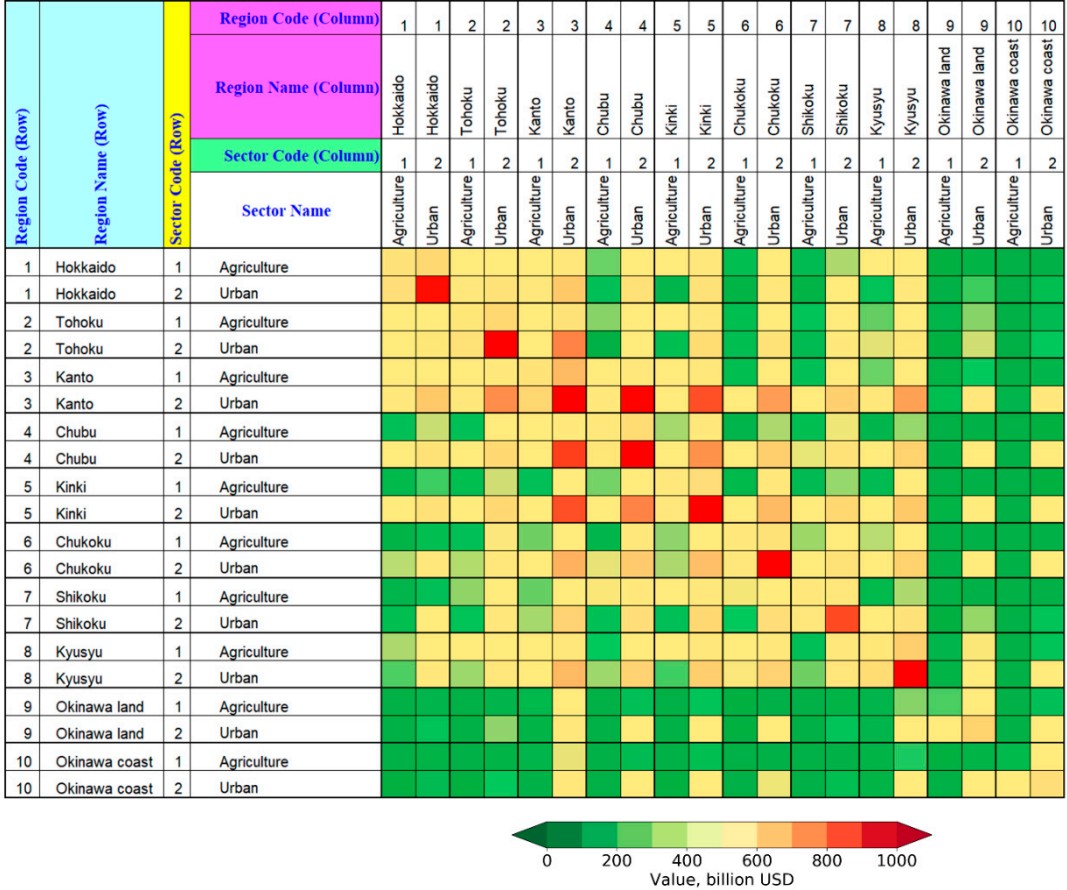

**Figure 10.** Multi-regional-input-output (MRIO) table originally established from this study and used to estimate the tsunami economic losses in the specific area.

**Table 4.** Total economic value and unit cost of the MRIO sector for Okinawa Island.

| Information | Land-Agriculture | Land-Urban | Coast-Agriculture | Coast-Urban |
|---|---|---|---|---|
| Total interaction, billion USD | 29.7 | 1465.5 | 14.1 | 727.8 |
| Value added, billion USD | 59.3 | 2553.6 | 32.8 | 1367.3 |
| Total economic value, billion USD | 89.0 | 4019.1 | 46.9 | 2095.1 |
| Area, sq.km | 195.4 | 165.1 | 47.1 | 70.9 |
| Unit cost, billion USD / sq.km | 0.455 | 24.343 | 0.995 | 29.550 |

Next, indirect economic losses are considered by the proposed algorithm. Figure 11 displays the economic losses that consist of direct (Figure 11a) and indirect (Figure 11b) losses, for which the indirect damage cost is approximately 56% that of the direct losses. Additionally, due to the direct losses, the indirect losses in the urban area of the coastal zone are the highest losses among the land use types. A comparison of losses with those in other regions is presented in Figure 12. The most damage in the Kanto area is approximately 8% of the direct losses. The second largest amount of damage is approximately 3% of the direct damage cost that occurs in the Kinki area, and next, the Kyusyu area has approximately 2.7% of the direct losses. For example, the indirect losses of urban regions in the coastal area of Okinawa is approximately 200 billion USD, which is distributed in the Kanto area of approximately 27.5 billion USD, which is the highest among the urban sector of seven regions excluding Okinawa. This indirect loss was distributed to the Kyushu area, at approximately 0.96 billion USD, which was the highest among the agricultural sectors of seven regions excluding Okinawa. This result means that Okinawa always has the closest relationship with Kanto in terms of tourism and industrial

technology. For the agricultural sector, the closest relationship is with Kyushu because it is the closest transportation hub.

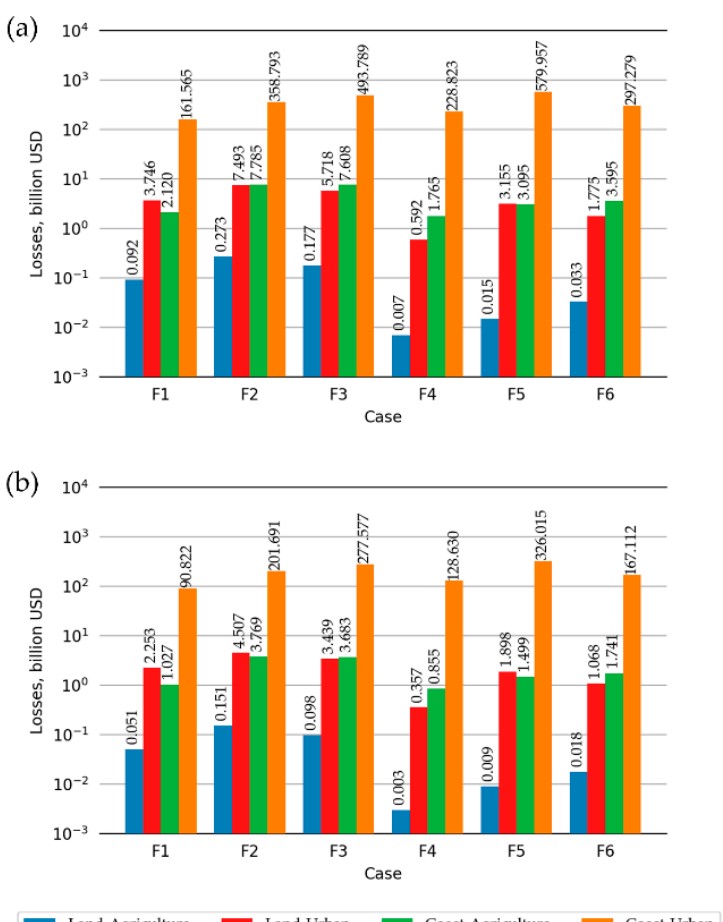

**Figure 11.** Economic losses in each tsunami case, (**a**) direct economic losses, (**b**) indirect economic losses.

A limitation of this study is the estimation of the damage in the same situation of loss (direct losses) in any flow depth, as mentioned in the methodology section, because we estimate the direct losses by overlaying the flood map with exposure (economic land use). For example, flood depths of 0.5 m and 2.0 m generate the same direct losses from the disaster, which may reflect an overestimation for the small flow depth and underestimation for the higher flow depth compared with the real situation. This problem that can be solved by using the fragility curve to identify the degree of damage from tsunami characteristics such as flow depth, flow velocity, and tsunami force [48,49].

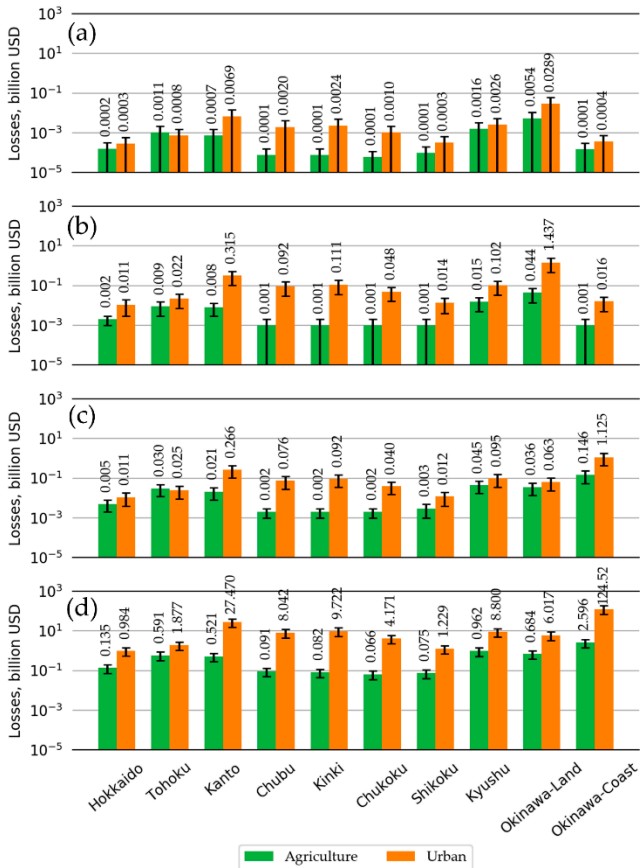

**Figure 12.** Impact of tsunami losses distributed in any region in Japan. (**a**) Impact of agriculture in inland areas, (**b**) impact of urban region in inland areas, (**c**) impact of agriculture in coastal areas, and (**d**) impact of urban region in coastal areas.

## 4. Conclusions

The aim of this study was to consider the level of tsunami damage caused by specific earthquake scenarios selected in this study on Okinawa Island, Japan, based on six earthquake scenarios. To estimate the spatial extent of tsunami flood damage, we used the TUNAMI-N2 model with a grid system from 810 m, 270 m, and 90 m. We determined the inundation area at 90 m resolution. The damaged areas in the six scenarios spanned 30 square kilometers. Our results showed that the tsunami flood area largely occurred in the urban areas, with inundation depths of 2.0–5.0 m.

An economic overview of nine regions in Japan represented by the IRIO table was modified by downscaling to the specific area (disaster area) using the topographic characteristic parameters, which produced the MRIO table. The downscaling was performed using the Chenery–Moses estimation method. The direct average loss of the tsunami damage scenario is approximately 350 billion USD, while the indirect loss is approximately 200 billion USD. In conclusion, direct losses will occur, and indirect losses will be approximately 56% that of the direct losses.

A limitation of the IRIO table (Figure 4) is that it cannot represent the economic value in a specific area, such as the disaster area, which can be solved by the MRIO table (Figure 5). The MRIO table was established by the Chenery–Moses estimation method to relate to the geographic characteristics (for dividing the coastal and inland areas), and part of the originality of this study is that it differs from the approach used by Pakoksung et al. [23]. We found that the MRIO table can be used to quantify the direct and indirect losses or trade in economic losses of tsunami disasters in each region and each sector. The main advantage of the MRIO table is the combination of a traditional MRIO table that allows policy-makers to clearly understand the path of a tsunami disaster based on the cost value. The MRIO

approach presented here is a descriptive device that is applicable to various regions and industrial sectors (urban and agriculture). It is a powerful tool for considering damage costs between regions.

The results of this study are limited by the inundation values estimated from the TUNAMI-N2 model, which uses a threshold of 0.3 m of water depth to estimate damage losses. To study tsunamis, there are several required variables, such as the travel time of the tsunami wave, its duration and peak, and the ranges of the inundation depths. As mentioned in previous research, building damage varies with flow depth, representing a fragility curve [17,18,50–52]. For cost estimations, the damage cost of the economic sector was estimated by the unit cost to demonstrate the economic direct losses, but these sectors have several other costs in real applications. In future work, the arrival time, duration, peak times and fragility curve can be used to identify the economic losses resulting from tsunami disasters. Finally, the results of this study can be used to evaluate tsunami disaster losses for tsunami protection and mitigation schemes, with a countermeasure methodology for future construction.

**Author Contributions:** K.P., A.S., K.M. and F.I. contributed to the implementation of the research, analysis of the results and writing of the manuscript.

**Funding:** This research was funded by the Willis Research Network (WRN) under the Pan-Asian/Oceanian tsunami risk modeling though the International Research Institute of Disaster Science (IRIDeS) at Tohoku University.

**Acknowledgments:** This study was conducted with the topography data provided by the Geospatial Information Authority of Japan (GSI). Economic data (IRIO table) were provided by Trade and Industry (METI), Japan. Earthquake scenarios were provided by Okinawa Prefectural Government. In this study, the QGIS software was used to illustrate the spatial data.

**Conflicts of Interest:** The authors declare no conflicts of interest.

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
