# Peer review of "Estimating Tsunami Economic Losses of Okinawa Island with Multi-Regional-Input-Output Modeling"

_geosciences, doi:10.3390/geosciences9080349_

Round 1

Reviewer 1 Report

The manuscript is well written, needing maybe a little grammatical editing (for example, page 4, line 182, I think the authors meant to say "finite difference" instead of finite different). Also, figure 9 is not readable at the scale/resolution it is presented, the authors may want to consider separating out those images. I feel the authors have a good handle on the tsunami modeling, per Dr. Imamura's research, but there needs to be more improvement on the economic modeling impact (whether using Input/Output methodologies or CGE). I think the authors should do a better job of reviewing the literature for the economic impact (direct/indirect) of disasters. There are other articles examining the economic impact of tsunamis (scenario or based on events such as the Tohuku-Oki earthquake and tsunami) to ports, trade, disruptions, damage locally (Okinawa), regionally and nationally.

Author Response

Dear Reviewer,

Thank you very much for taking the time to read my manuscript. The responding is in the attachment.

Reviewer 2 Report

I think the paper should be published

I have the following comments:

In the abstract cut the last sentence starting with “we hope to apply...” you may use this sentence in the discussion but not in the abstract

Please shorten the paragraph of the examples of use of TUNAMI-N2,two or three examples are enough. 

Introduce a short descriptions of the impact of past tsunamis(presented in figure in 1) in the island with inundation and run up

Concerning the flow depths and flux velocity I would like to see the comparison of your numbers with field data. I am not saying they are wrong but there were so many field surveys after this event that I believe this w comparison with field data would improve the manuscript and give more confidence to apply it in other areas not only in Japan but in any part of the world

Line 228 explain in detail what values are used to differentiate the coastal area: altitude limit, topography gradient limit and distance from the shore - this will help others to apply your methodology to other areas

Correct figure3 caption it should be “was selected” instead of “ was collected”

Figure 4 is unnecessary . Tsunami modelers know what is the nesting system. However, You should mention in the manuscript what is bathymetry data source. Is it GEBCO in open ocean? And for the inundation computation what data is used? 

Table 2 needs a more detailed explanation please explain how you get itp

Figure 3 caption explain what the color rectangles represent ( I presume they are the fault areas, but are the size of the rectangles compatible with dimensions given in the earthquake parameter table?

What is the use of figure 5a? I believe none....better take it out.

Reading through the conclusions paragraph we get the feeling that your model still needs a lot of improvement, better rephrase it and show how TTT. Fragility curves etc in your methodology would give different results or not and insert this text in the discussion paragraph or merge the two “Discussion and Conclusions” 

Author Response

(The authors gave the same response as above.)

Reviewer 3 Report

The article "Estimating Tsunami Economic Losses of Okinawa Island with Multi-Regional-Input-Output-Modeling" illustrates a methodology to estimate economic losses caused by specific tsunami scenarios in Okinawa Island. The methodology uses a tsunami propagation model and an economic model.

I think, there is a relevant issue which needs to be attended by the authors. The earthquake-scenario-based approach adopted by the authors to assess the tsunami hazard is too simplistic. Many papers have shown that constant slip faults, such as those of this paper, are associated with significantly smaller tsunamis, as compared to those simulated with more realistic spatially-varying slip (e.g. Geist & Dmowska (1999), Li et al. (2016), Sepulveda et al. (2017)). On the other hand, it seems to me that they are not working with the worst-earthquake-scenarios in these faults (for instance, a greater earthquake merging more than one segment in Okinawa Trough and Ryukyu Trough may occur). The study needs to include more references to support the selection of these six earthquake scenarios or improve this important step in the methodology (e.g. by considering a Probabilistic Tsunami Hazard approach or use worse scenarios).  It is important to tell the reader that the simulated tsunamis (and so economic losses) may be significantly underestimated in this study.

I also recommend to clarify what the authors did in the past. For instance, what is the difference and new contributions with respect to Pakoksung et al. (2017), “Approach of Estimating Tsunami Economic Losses in the Okinawa Island with Scenario-Based of Input-Output Table and Okinawa Earthquake Sources”. I reviewed that paper briefly and I could find very similar procedures as those presented in the new paper. I recommend to compare both studies and check the novelty of the new paper.

Here I give some specific recommendations to each section of the paper. I also attached a pdf with notes.

Abstract: First, I think the abstract is too long. It gives too many details about some aspects, but insufficient details about others (e.g. what “land cover” means?). I recommend to re-write the abstract. Second, there are very similar sentences to those written in the Introduction section (e.g. Tsunamis, as major natural hazards, can affect human lives and cause major economic…” v/s “Tsunami, one of Earth’s major natural hazards, affects human life and property”). Those general aspects are not necessary to be written in the abstract.

1. Introduction: 

-I think the first paragraph is fine because it briefly shows the consequences after the 2011 Japan tsunami and highlights the importance of mitigation strategies.

-The sub-section “Tsunami model” contains too many elements which are not relevant for the paper. Since this is a small paper, text should focus on main aspects. For instance, no need to name three models (maybe you can give citations only). Perhaps you can focus on Tunami-N2 only and describe it briefly. The illustration cases (2004 and 2011) are unnecessary. Values reported for these events about flow depths and velocity are site-specific (i.e., not general) and do not offer any relevant information to your study.

-The sub-section “Economic model” is better, but it also gives too many details about illustrations which are not related with the present study (e.g. description of IO model for water). I recommend to focus on Pakoksung et al. (2017) and also indicate what is new and what is repeated.

2. Materials and methods: The first paragraph of this section repeats some statements mentioned in other sections.

-Section 2.1 mentions that earthquakes are modeled with a constant-slip and a rectangular fault. This simplistic model has been proven to underestimate the tsunami for a given earthquake magnitude. This is a relevant issue of the study.

-Section 2.2. The tsunami model is described three times in the paper (do it a single time!). I don’t think you need to write and define the shallow water equations because many papers did this already (even for TUNAMI-N2). You can just mention and cite these papers. On the other hand, the paper needs to give details about two important aspects:

a) The wave runup model commonly needs a special moving boundary scheme. The runup model has to be described because all your results rely on this (the runup and inundation at the coast).

b) You need to explain what Manning number are you using for your bottom friction model and how different values can affect your results (note in Bricker et al. (2015) that Manning numbers can vary in one order of magnitude). Perhaps you may need a sensitivity analysis of the economic losses to the model uncertainties.

-Section 2.3. The presentation of this section is fine. I would only erase some irrelevant iniformation.

3. Results and Discussion: Need to check definitions (e.g. T.P.).

4. Conclusions: Several statements appear for the first time in this section. Present these statements earlier in the text and briefly summarize your findings here.

Conclusively, I think the paper is very relevant for illustration purposes because it shows how tsunamis and economic losses can be related. However, the authors should (at least) aware the reader about the simplicity of the tsunami hazard assessment. Finally, I also attached a pdf indicating some typos and further recommendations.

References:

Geist, E. L., & Dmowska, R. (1999). Local tsunamis and distributed slip at the source. In Seismogenic and tsunamigenic processes in shallow subduction zones (pp. 485-512). Birkhäuser, Basel.

Li, L., Switzer, A. D., Chan, C. H., Wang, Y., Weiss, R., & Qiu, Q. (2016). How heterogeneous coseismic slip affects regional probabilistic tsunami hazard assessment: A case study in the South China Sea. Journal of Geophysical Research: Solid Earth121(8), 6250-6272.

Sepúlveda, I., Liu, P. L. F., Grigoriu, M., & Pritchard, M. (2017). Tsunami hazard assessments with consideration of uncertain earthquake slip distribution and location. Journal of Geophysical Research: Solid Earth122(9), 7252-7271.

Pakoksung, K., Suppasri, A., & Imamura, F. Approach of Estimating Tsunami Economic Losses in The Okinawa Island with Scenario-based of Input-Output Table and Okinawa Earthquake Sources. Internet J. of Society for Social Management Systems11(1).

Bricker, J. D., Gibson, S., Takagi, H., & Imamura, F. (2015). On the need for larger Manning's roughness coefficients in depth-integrated tsunami inundation models. Coastal Engineering Journal57(02), 1550005.

Author Response

(The authors gave the same response as above.)

Round 2

Reviewer 3 Report

Thank for consider the corrections. I think it is important to point out all the potential limitations and assumptions of the study, so future user are aware of them.